# Arabian Sea upwelling over the last millennium and in the 21st century as simulated by Earth System Models

Xing Yi and Eduardo Zorita

Helmholtz-Zentrum Geesthacht, Institute of Coastal Research, Max-Planck-Str.1, Geesthacht, 21502, Germany

Correspondence to: Xing Yi (email: xing.yi@hzg.de)

**Abstract.** Arabian Sea upwelling in the past has been generally studied based on the sediment records. We apply two earth system models and analyse the simulated water vertical velocity to investigate the variations of the coastal upwelling in the western Arabian Sea over the last millennium. In addition, the same models, with slightly different configurations, are also employed to study the changes in upwelling in the 21st century under the strongest and the weakest greenhouse gas emission

- scenarios. With a negative long-term trend caused by the orbital forcing of the models, the upwelling over the last millennium is found to be closely correlated with the sea surface temperature, the Indian summer Monsoon and sediment records. The future upwelling under the Representative Concentration Pathway (RCP) 8.5 scenario reveals a negative trend, in contrast with the positive trend displayed by the upwelling favourable along-shore winds. Therefore, it is likely that other factors, like water stratification in the upper ocean layers caused by the stronger surface warming overrides the effect from
- the upwelling favourable wind. No significant trend is found for the upwelling under the RCP2.6 scenario, which is likely due to a compensation between the opposing effects of the increase in upwelling favourable winds and the stratification of the water column.

#### **1** Introduction

Upwelling lifts the cold nutrient-rich water from deeper ocean layers to the surface, which cools the surface water and 20 provides the biologically active layers with nutrients. It has a great impact on human activities. For instance, about 20% of the total marine fish catches originate from upwelling regions that cover only 2% of the entire ocean area (Pauly and Christensen, 1995). Upwelling also affects the climate by cooling the sea surface temperature (SST) and by influencing the air-sea interactions associated with the variation of SST (Izumo et al., 2008). It has been suggested that, under the global warming scenario, coastal upwelling at global scale would be intensified as the upwelling favourable wind-stress would be

strengthened due to the enhanced air-sea temperature gradient (Bakun, 1990). Numerous studies have provided insights into the upwelling variations over the last few decades. Most of these studies focuses on the four major eastern boundary upwelling systems (EBUSs), namely, the California (Schwing and Mendelssohn, 1997), Canary (McGregor et al., 2007), Humboldt (Gutiérrez et al., 2011) and Benguela (Santos et al., 2012) upwelling systems. In support of the Bakun hypothesis, Narayan et al. (2010) found positive trends over the late 20th century in all these four coastal upwelling regions and

Sydeman et al. (2014) also reported upwelling intensification in the major EBUSs after synthesizing the results from previous studies.

In addition to the major EBUSs, the Arabian Sea is one of the most productive regions in the world. The coastal upwelling in the western Arabian Sea is mainly driven by the southwest (SW) wind-stress which is induced by the Indian summer Monsoon (ISM). Since the link between the ISM and the upwelling is very pronounced, many studies have used the ISM as an indicator of the upwelling, and vice versa. However, direct observations of upwelling in terms of water mass vertical velocity over a long time period are rare, thus, alternative upwelling proxies such as SST, surface wind-stress and sediment records have been generally applied. An alternative tool to analyse upwelling variability is provided by model simulations.

The knowledge of upwelling evolution in the past is crucial to understand and model the upwelling variations at present and to predict them in the future. A widespread approach to study upwelling in the past is through sediment records where the abundance of *G.bulloides* is stored. *G.bulloides* belongs to the phylum foraminifera and is very sensitive to upwelling variations so it is generally used as an upwelling proxy. Over the last millennium the Arabian Sea upwelling is reported to

- exhibit a slight decrease until approximately 1600 and an abrupt increase afterwards (Anderson et al., 2002; Feng and Hu, 2005; Sinha et al., 2011). As for the future, Wang et al. (2015) used the Coupled Model Intercomparison Project Phase 5 (CMIP5) model outputs to conduct an analysis on the 21st century upwelling under the future scenario of Representative Concentration Pathway (RCP) 8.5. They indicated that the upwelling intensity and duration are both increased at high latitudes in most of the major EBUSs except California. However, a study focused on the future upwelling in the Arabian
- Sea is not yet available.

We investigate here the Arabian Sea upwelling variation over the last millennium by analysing model simulations of water vertical velocity and by comparing the model results with the observational sediment records to determine the existence of the long-term trends. The evolution of future upwelling in the Arabian Sea is also investigated by using the same models (albeit with a slightly different configurations for past and future). We compare the future upwelling under the RCP8.5 and

25 (albeit with a slightly different configurations for past and future). We compare the future upwelling under the RCP8.5 and RCP2.6 scenarios to gain the information on how the greenhouse gas emission level could affect the variation of upwelling.

#### 2 Model and data

In this study, we analyse the outputs of two earth system models: the Earth System Model of Max-Planck Institute for Meteorology (MPI-ESM) (Giorgetta et al., 2013) and the Community Earth System Model (CESM) (Hurrell et al., 2013). For the analysis over the last millennium, we use the paleo configuration of the MPI-ESM (MPI-ESM-P) and the Last Millennium Ensemble (LME) project (Otto-Bliesner et al., 2016) of the Community Atmosphere Model Version 5 from

CESM (CESM-CAM5). To estimate the upwelling variabilities in the future, we apply the low resolution configuration of the MPI-ESM (MPI-ESM-LR) and the Community Climate System Model (CCSM) which is the predecessor of the CESM.

The MPI-ESM-P and the MPI-ESM-LR share the same horizontal resolutions of about 2 degrees for the atmosphere
(192×96) and about 1 degree for the ocean (256×220). Their vertical resolutions are the same as well with 47 levels for the atmosphere and 40 levels for the ocean. In spite of this, the MPI-ESM-P is available for the last millennium (850-1849) and the MPI-ESM-LR covers the future period (2006-2100) simulated under the greenhouse gas emission scenarios. We analyse an ensemble of three simulations which have almost the same external forcings but started with different initial conditions. The CESM-CAM5 and the CCSM4 also have identical horizontal resolutions for the ocean which is about 1 degree on the latitude (320×384). For the atmosphere, the CESM-CAM5 has a horizontal resolution of 2.5

- degrees on the longitude and 1.875 degrees on the latitude (144×96), which is twice coarser than the CCSM4 (288×192). On the vertical, they share the same resolutions with 60 levels in the ocean and 30 levels in the atmosphere. The CESM-CAM5 covers the last millennium period (850-1849) and the CCSM4 covers the 21st century (2005-2100). Each of them also has an ensemble of many simulations that are forced by identical external forcings but different initial conditions. In order to be
- comparable with the MPI-ESM models, we select three simulations each from the CESM-CAM5 and CCSM4 ensembles.

The differences between simulations within each ensemble provide an estimation of the amplitude of the effect of internal climate variability, whereas the signal shared by all simulations will approximate the response to the imposed external forcing (Tim et al., 2016).

We investigate several variables that are modelled by the simulations, including upwelling velocity, sea surface temperature (SST), surface wind-stress, wind speed, and sea level pressure (SLP). The upwelling velocity is the "vertical velocity" in the CESM ensembles but has to be derived from the "vertical water mass transport" for the MPI-ESM ensembles. Since coastal upwelling in the western Arabian Sea occurs from 200 meters below the ocean surface (Brock and McClain, 1992), we average the upper 200 meters of the data. We use the monthly data from the models and because the upwelling season starts

average the upper 200 meters of the data. We use the monthly data from the models and because the upwelling season starts in May and ends in September (Brock et al., 1991), only the summer months June, July and August (JJA) are selected. One exception is that for the SST we choose July, August and September (JAS) due to the lag in the response of SST to the upwelling (Rixen et al., 2000). In addition to the modelled data, we also apply the sediment records used by Anderson et al. (2002) to compare with our results.

# 30 3 Arabian Sea upwelling in the last millennium

The mean summer upwelling velocities in the Arabian Sea modelled by MPI-ESM-P and CESM-CAM5 for the last millennium are presented in Fig. 1. To avoid duplication, we only show the results of one simulation from each of these two

simulation ensembles since the three simulations in each ensemble share very similar spatial patterns. The models present identical patterns of the summer upwelling in the Arabian Sea where strong upwelling occurs along the coast especially in the western Arabian Sea, induced by the southwest wind-stress (Fig. 1a and 1b). Coastal upwelling in the western Arabian Sea is more intense in the MPI-ESM-P simulation, where the velocity can reach 1.5 m/day, than in CESM-CAM5 where the maximum velocity is around 1 m/day. The magnitude of these velocities is reasonable as it matches the estimated order of magnitude suggested from studies focused on present time period (Rixen et al., 2000; Shi et al., 2000). In addition, weak downwelling in the central Arabian Sea is found in both models. Downwelling here results from the convergence zone generated by the wind-stress curl (Thadathil et al., 2008; Bauer et al., 1991). Thus, the spatial distribution of the vertical velocity is consistent with the Ekman pumping effect (Lee et al., 2000).

5

We average the upwelling along the coast and calculate the upwelling velocity time series of this area (Fig. 1c and 1d). In general, these time series show that on average the mean upwelling velocity modelled by MPI-ESM-P is larger than the one simulated by CESM-CAM5 by around 0.3 m/day. The multidecadal variation is, however, larger in CESM-CAM5. All six simulations by the two models reveal negative trends of the upwelling velocity although not all of them are significant at the 95% significance level. A detailed interpretation of these trends will be presented in the "Upwelling trends" section.

Since upwelling lifts the cold water from the deeper layers to the ocean surface during the upwelling season, the SST is often anticorrelated with upwelling in the upwelling region. This relationship is well captured in both of the models (Fig. 2a and 2b). Strong negative correlations are shown along the coastal upwelling regions and even greater (r

As a comparison with the sediment records which is considered as the observational data, Fig. 2c and 2d show the correlations between Arabian Sea upwelling and the *G.bulloides* abundance retrieved from the sediment cores RC2730 and RC2735 (Anderson et al., 2002). This record has been interpreted as an indicator of upwelling in this region (Peeters et al., 2002). We only mark the location of RC2730 on the maps because these two cores are located very close to each other. The

- calculation of the correlation is performed on the 50-year averaged data during the overlapping years (1050-1849) of our modelled upwelling velocity and the *G.bulloides* abundance. These filtered series presumably reflect the variations in the external forcing, or at least the externally forced component of the variability should be large. Since the external forcing should be ideally the same in the observations and in the simulations, a positive correlation between both records should be expected. However, the records from only two cores located at very close positions might not represent the upwelling in a
- broader area. In spite of this, the maps present significant correlations along the coast all the way to the northern Arabian Sea especially for the MPI-ESM-P model. Such correlations indicate that the models reasonably reproduce the variability of upwelling velocities at time scales that are presumably driven by the external climate forcing. The simulated records are thus comparable to the sediment records.

#### 4 EOF analysis of upwelling

- We perform an Empirical Orthogonal Function (EOF) analysis (von Storch and Zwiers, 2001) on the Arabian Sea upwelling to identify its main spatial variation patterns and the corresponding temporal evolutions. EOF analysis can identify the spatial patterns that describe the data variance by generating the EOF modes and their corresponding principal component (PC) time series, where each mode is ranked by its explained proportion of the total variance.
- The leading two modes from the EOF analysis of the upwelling are given in Fig. 3. The ranks of the first two modes are switched between MPI-ESM-P and CESM-CAM5. The first mode from CESM-CAM5 (Fig. 3b) accounts for 41% of the total variance and reveals similar spatial patterns as the second mode of the MPI-ESM-P simulation (Fig. 3c) which accounts for 24% of the total variance. We find that these EOF modes are related to the interannual variations as they capture the spatial patterns of the mean state of the upwelling (Fig. 1a and 1b) where the different signs between coastal and central
- Arabian Seas show the contrast between these regions. Their PC time series (Fig. 4b and 4c) also have a strong positive correlation with the time series averaged from the coastal upwelling in the western Arabian Sea (Fig. 1c and 1d) for all the simulations respectively (Table 1).

Thus, the intensity of the coastal upwelling in the western Arabian Sea is in phase with the intensity of the upwelling in the 30 rest coastal areas and also with the intensity of the downwelling in the central Arabian Sea. This result is supported by the correlation between upwelling and the Indian Monsoon Index (IMI). We calculate the IMI from the model derived U850

wind data based on the definition of Wang and Fan (1999). The IMI is significantly correlated to our upwelling time series in all the simulations (Table 1).

It is shown in Fig. 5a and 5b that the IMI is also correlated to the upwelling velocity in the rest coastal areas in the Arabian 5 Sea and the negative correlation in the centre downwelling region indicates that the IMI contributes to the intensification of the downwelling as well. Since the IMI is calculated based on the wind field which is caused by the sea level pressure (SLP) gradient, we also apply the EOF analysis to the SLP field. We find that the time series of the coastal upwelling in the western Arabian Sea from both of the models are also significantly correlated with the PC time series of the second mode from the EOF analysis of the SLP (Table 1).

10

This EOF mode of SLP displays a clear spatial pattern representing the contrast between Africa and Indo-Asia (Fig. 5c and 5d). In general, the patterns revealed from the two models are very similar. However, the boundary line that separates the positive and negative EOF phases rotates anticlockwise in CESM-CAM5 comparing to that in MPI-ESM-P. This tilt might be responsible for the 8% more variance captured by the second EOF mode of SLP from CESM-CAM5 than that from MPI-

15 ESM-P as well as the higher correlation between the SLP PC2 and the coastal upwelling time series derived from CESM-CAM5 than that from MPI-ESM-P (Table 1). An explanation might be that, with this tilt, the second EOF mode revealed from CESM-CAM5 captures a contrast of the spatial patterns in the southern Arabian Sea, which contributes to the variance representing the SW wind-stress that highly correlates with the time series of the coastal upwelling in the western Arabian Sea.

#### 20 **5 Upwelling trends**

In order to understand the millennial scale variability of the Arabian Sea upwelling, we calculate the long-term linear trends of the vertical velocity. Figure 6a and 6b show the upwelling trends derived from the two simulations. Both models reveal negative trends in the northern Arabian Sea and along the coast where the intense upwelling occurs. On the contrary, the central and eastern Arabian Seas display positive trends. Note that the central Arabian Sea is dominated by downwelling, and

- so the positive trends in this region actually indicate a weakening of downwelling, whereas in the eastern Arabian Sea the upwelling is slightly strengthened. However, in the western Arabian Sea, the region of our main focus, the upwelling velocity decreased over the last millennium, whereby the MPI-ESM-P model displays a more significant reduction than the CESM-CAM5 model. The upwelling velocity trends share very similar spatial patterns with the first mode of MPI-ESM-P (Fig. 3a) and the second mode of CESM-CAM5 (Fig. 3d) from the EOF analysis. In addition, their PC time series (Fig. 4a)
- 30 and 4d) also confirm the trends revealed in the upwelling velocity time series (Fig. 1c and 1d). The different signs in the EOF spatial patterns and the trends in the PC time series indicate the weakening of the western Arabian Sea coastal upwelling. The trends revealed from the time series are small in terms of their amplitudes over a thousand years compared to

the mean upwelling velocity. However, negative trends of western Arabian Sea upwelling are shown consistently in all the six simulations by the two models despite of the different internal conditions used for the simulations. Therefore, the weakening of upwelling in this region is very likely a robust feature in the simulations.

5 These trends are likely induced by the external forcing used to drive the models. Among all the external forcings, only the orbital forcing displays the long-term millennial scale trend, so that the identified upwelling trends can presumably be attributed to the orbital forcing.

To confirm the effect of the orbital forcing, we calculate the trends of the SLP (Fig. 6c and 6d) and the SW wind-stress (Fig. 6e and 6f) since the upwelling is mainly forced by the SW wind-stress which is generated from the SLP gradient. The spatial patterns of the SLP trends show clearly that in mid latitudes the SLP tends to increase and in low latitudes the SLP tends to decrease. These patterns are quite likely resulting from the effect of orbital forcing, as similar long-term changes are also derived from the differences between equilibrium mid-Holocene and present climate simulations (Braconnot et al., 2002). With positive trends in the mid latitudes and negative trends in the low latitudes, the SLP contrast between these areas is

- 15 reduced, which further affects the wind in the Arabian Sea. During the summer upwelling season, this SLP contrast drives the SW wind so when the SLP contrast is reduced the SW wind is also weakened. Figure 6e and 6f present the negative trends of SW wind-stress as expected. In general, the trend from MPI-ESM-P is larger than that from CESM-CAM5. Since the western Arabian Sea coastal upwelling is significantly linked to the SW wind-stress in the Arabian Sea, the negative trends of SW wind-stress can cause the decrease in the upwelling velocity (Fig. 6a and 6b). A larger trend of SW wind-stress
- 20 in the case of the MPI-ESM-P model than in the CESM-CAM5 model also results in stronger trend of upwelling. Therefore, the weakening of the coastal upwelling in the western Arabian Sea is induced by the reduction of the SW wind-stress which results from the long-term change of the SLP contrast between mid and low latitudes due to the orbital forcing.

#### **6** Future scenarios

In addition to the analyses of the last millennium simulations, we study the trends in the future scenarios as well. Figure 7 presents the time series of the SST, the SW wind-stress and the upwelling velocity modelled by MPI-ESM-LR and CCSM under the scenario RCP8.5.

All the simulations by the two models are quite consistent regarding the results for the RCP8.5 scenario by showing positive trends in SST and SW wind-stress but negative trends in upwelling velocity. The SST (Fig. 7a and 7b) increases significantly under the effect of the greenhouse gas emission. In the MPI-ESM-LR model (Fig. 7a) the SST begins with lower values than that in the CCSM model (Fig. 7b) but they rise to similar values in the end of the simulations so the SST modelled by the MPI-ESM-LR has a lager trend than by the CCSM. The SW wind-stress (Fig. 7c and 7d) is also strengthened due to the

enhanced contrast between the surface heating over the land and the sea. Although all the simulations by both models show positive trends, the CCSM simulations (Fig. 7d) reveal more distinct and significant trends than the MPI-ESM-LR (Fig. 7c). As a result of the intensification of the wind-stress the upwelling velocity is expected to increase as well. However, negative trends of upwelling velocity are found in all the simulations (Fig. 7e and 7f) especially for the MPI-ESM-LR (Fig. 7e). This raises an interesting question as to what is causing the negative trend in upwelling velocity given that the upwelling favourable wind-stress is projected to increase. Figure 7 shows that the MPI-ESM-LR simulations reveal larger positive trends in SST and at the same time larger trends in upwelling than the CCSM simulations, which implies that the drop in upwelling velocity might be linked to the warming of the upper ocean layers in the RCP8.5 scenario and thus to an increase in stratification. To test this hypothesis, we perform the same analysis for the RCP2.6 scenario (Fig. 8).

10

5

Under the RCP2.6 scenario the SST still increases in both models (Fig. 8a and 8b) but not as much as in the RCP8.5 scenario. However, no consistent trends for all the simulations are found in the SW wind-stress (Fig. 8c and 8d) or the upwelling velocity (Fig. 8e and 8f). Thus, it is very likely that the lack of negative trends in upwelling velocity under scenario RCP2.6 is caused by the negligible ocean surface warming. Note that the correlation between upwelling and SW wind-stress at interannual timescales does not change much in different scenarios (Table 2), which indicates that the

15 wind-stress at interannual timescales does not change much in different scenarios (Table 2), which indicates that the influence of the wind-stress on the upwelling remains at the same level in the two scenarios.

The correlation between upwelling and the SW wind-stress is slightly weaker in the RCP2.6 than in the RCP8.5 scenarios from the results of the MPI-ESM-LR but stronger from the CCSM. However, the significant positive correlation between upwelling velocity and the SW wind-stress does not lead the evolutions of upwelling and wind-stress towards the same direction under a stronger greenhouse gas emission scenario (Fig. 7). One explanation for this is that even though the correlation between upwelling and the SW wind-stress remains at the same level in the two scenarios, the significant rise in the SST under the RCP8.5 scenario might override the effect of the wind-stress. Although wind-stress still drives upwelling, its effect is obstructed by the more stable water column due to surface warming effect. This is confirmed by Fig. 9, which shows the vertical trends and means of water temperature and upwelling velocity modelled by the two models for the RCP8.5 scenarios.

In addition to Fig. 7 and Fig. 8, the differences in the trends at all levels in the upper 200m between the RCP8.5 and RCP2.6 scenarios are shown in Fig. 9. There are slight positive trends for RCP2.6 and significant trends for RCP8.5 in the water

30 temperature (Fig. 9a and 9c). There are also no consistent trends in the upwelling for RCP2.6 but significant negative trends at all levels for RCP8.5 (Fig. 9b and 9d). In the upper 50m to 100m the upwelling velocity show larger negative trends when the trends of the water temperature increase which is very clearly shown in the CCSM plots (Fig. 9c and 9d). At around 50m depth the temperature trends reach their peaks and the upwelling trends also get to their lowest values (in the surface layer above 100m). The upwelling is actually most intense at this layer (Fig. 9d) but it is most reduced as well, which causes the

drop in the upwelling time series shown in Fig. 7. Therefore, in spite of the positive correlation between the SW wind-stress and the upwelling, the warming of the upper ocean layers under the RCP8.5 scenario reduces the upwelling velocity more efficiently.

#### 7 Discussion

- 5 One limitation of this study is that it relies to a great extent on the realism of the models employed, which makes it very difficult to validate the results as there are no available direct observations on upwelling over the time span. However, we do find a connection between our modelled upwelling velocity and the observational sediment records. They present a significant correlation (Fig. 2) which indicates the effect of the external forcing. In addition, they also have similar patterns in the time series in terms of the trends. It can be noticed that the long-term trend revealed from the upwelling time series 10 (Fig. 1) tends to flip, especially in the CESM-CAM5 model, at around the year 1550. This flip was reported by Anderson et
- al. (2002) where they reconstructed the upwelling index (monsoon winds) from the sediment records over the last millennium. They found a slightly negative trend from 1000 to 1500 and a positive trend thereafter, which is very similar to our finding. Thus, we divide our upwelling time series into two periods based on the flip and calculate the trends separately (Table 3).

It is very clear that five out of six of the simulations indicate a flip in the long-term trend at 1550 as the signs of the trends change from negative to positive after this point (except for the CESM-CAM5 r1). Therefore, our models are able to produce realistic upwelling velocity time series that agree with the sediment records but there are still remaining uncertainties. For instance, the exact time when the trends change signs varies from simulation to simulation, which might be caused by the different initial conditions applied for each simulation.

Model resolution also has an impact on the realism of the results (Small et al., 2015; Ranjha et al., 2016). However, it is still unclear what resolution is most suitable to model the Arabia Sea coastal upwelling system. In our study, the MPI-ESM uses a coarser horizontal resolution (256×220) than the CESM (320×384) and produces the upwelling velocity with larger means

- but less variability, as well as larger trends. The upwelling velocity simulated by the CESM, however, has stronger correlation with the wind and the wind based IMI. Whether these differences are caused by the different resolutions is uncertain but this comparison might shed light on how the modelled Arabian Sea upwelling response to the change in the model resolutions.
- A recent study found a reduction in the future upwelling at low latitudes in most of the four EBUSs (Wang et al., 2015). In our analysis for the future scenario RCP8.5, the upwelling velocity also reveals a negative trend whereas the SW wind-stress presents a positive trend. Thus, the reduction of the upwelling is not caused by the wind-stress, although they are still

significantly correlated. The analysis for the RCP2.6 scenario does not show a trend in either the upwelling or the SW windstress but it shows strong correlation between upwelling and the wind-stress as well as the RCP8.5 scenario. Wang et al. (2015) suggested several mechanisms that could lead to a reduction in the upwelling at low latitudes such as strengthened downwelling favourable winds, weakened upwelling favourable winds, and enhanced water stratification due to the greenhouse warming. In our case, it is the stratified water under the RCP8.5 scenario that reduces the upwelling but this is not shown in the RCP2.6 scenario. Therefore, it remains for further study to investigate at what level of future emissions between the strongest and the weakest warming scenarios water stratification and the effect of wind-stress approximately balance.

#### 8 Conclusion

- The variability of the Arabian Sea upwelling over the last millennium and in the 21st century are investigated in this study. We statistically analyse and compare the outputs of two earth system model simulation ensembles and conclude:
  - Over the last millennium, the modelled upwelling velocity has a strong correlation with the SST and the IMI as well as the observed *G.bulloides* abundance. This indicates a close connection among the Arabian Sea upwelling, the Indian summer Monsoon and the sediment records.
  - A negative long-term trend is found in the upwelling velocity over the last millennium, which is resulting from the orbital forcing of the models. In the last 300 years, however, the upwelling reveals a positive trend, which matches the observations in the sediment records.

15

• The correlation between upwelling and the SW wind-stress remains at the same level under the RCP8.5 and RCP2.6 scenarios in the future but the stronger greenhouse gas emission scenario leads to opposite trends in the SW wind-stress and the upwelling velocity. The positive trend in the SW wind-stress is caused by the intensified SLP gradient between land and ocean due to the warming effect. The upwelling velocity shows a negative trend because under the stronger scenario, the warming of the sea water tends to stabilize the surface layer, which overrides the effect of the SW wind-stress and interrupts the upwelling.

### Acknowledgements

This work is funded by the Cluster of Excellence Integrated Climate System Analysis and Prediction (CliSAP) Project B3. We thank the Max-Planck-Institute for Meteorology for providing the MPI-ESM model data. All the other publicly available data used in this study are gratefully acknowledged.

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
