# Peer review of "Arabian Sea upwelling over the last millennium and in the 21st century as simulated by Earth System Models"

_Climate of the Past, 2016_

## Short Comment (SC1) · 18 Dec 2016

Arabian Sea upwelling, temperatures and monsoon activity have been documented to show major natural variability and cyclicity. It is therefore commendable that the authors attempt to calibrate and verify their model against pre-industrial natural changes of the past 1000 years before embarking on the forward modelling part, the future predictions. In chapter 3 of their manuscript, the authors say that this hindcast was successful, therefore the model can be used for simulations of future upwelling. There are, however, two issues that cast doubt on the validity of this hindcast calibration:

1) DOES THE MODEL USED IN THIS STUDY ACCOUNT FOR SOLAR ACTIVITY FORCING, WHICH IS KNOWN TO BE STRONG IN THE ARABIAN SEA?

[Figure]

Holocene upwelling, temperatures and monsoon activity in the Arabian Sea are long known to be controlled significantly by changes in solar activity changes. There is a great amount of literature on this subject, see e.g. the recent paper by Munz et al. 2016 http://www.clim-past-discuss.net/cp-2016-107/ Other important papers: Neff et al. (2001), Gupta et al. (2005), Thamban et al. (2007), Menzel et al. (2014). You can find the detailed references in the regional synthesis in Lüning & Vahrenholt 2016 (page 296) https://www.researchgate.net/publication/308928345_The_Sun%27s_Role_in_Climate

It is clear that any model used for climate simulations in the region has to account for this strong solar forcing that has been clearly documented in many studies from the region. In order to fully calibrate the model, it would be important to extend the calibration period backwards several millennia. Only then, the millennial-scale cyclicity becomes apparent.

2) USE UPDATED AND LONGER VERSION OF UPWELLING PROXY RECORD

The authors are using the uwpelling reconstruction of Anderson et al. 2002 as comparison for the hindcast comparison. However, this paper has been superseded by a much improved and longer version published in Anderson et al. 2010, uncited by Yi & Zorita. http://onlinelibrary.wiley.com/doi/10.1002/jqs.1369/abstract

The age model has somewhat changed and the dataset been extended. It is crucial that the model used in Yi & Zorita replicates e.g. the weak monsoon phase during the 'Dark Ages Cold Period' described by Anderson et al. 2010:

"In addition to the well-known decrease in the Indian summer monsoon since the early Holocene 9 ka BP, we found a minimum in the monsoon 1500 a BP, following by an increasing trend to the present. Superimposed on this increasing trend is a strong monsoon interval 1200–800 a BP and a weak monsoon interval 600–400 a BP that we correlated with the Medieval Warm Period and Little Ice Age in a previous study (Anderson et al., 2002)."

Other proxy series in the region that confirm the strong climate variability and cyclicity in the Arabian Sea are: Van Rampelbergh et al. 2013: http://www.sciencedirect.com/science/article/pii/S0277379113000255

Hoti Cave: Fleitmann 2008: http://www.speleothem2013.uni-hd.de/materials/S4_Fleitmann.pdf Fleitmann et al. 2007: http://www.sciencedirect.com/science/article/pii/S0277379106002265

---

## Referee Comment (RC1) · Anonymous Referee #1 · 22 Dec 2016

General comments: The authors discuss Arabian Sea upwelling based on Earth System model simulations of the past and future projections using various forcing/GHG scenarios. Results are presented based on statistical analyses of model outputs from the MPI-ESM and CESM models. A key result of the study is that stratification in the upper Arabian Sea caused by stronger warming under GHG scenarios has the potential to override the effect from upwelling favourable winds. The manuscript should be published once my comments have been addressed.

Details: 1. Page 3, line 11: ". . . with an area per grid point which is four times coarser . . . You are referring here to an area i times j grid points, i.e. the area per model grid point in CESM-CAM5 is four times coarser than in CCSM4.

2. Page 4, last paragraph: although the conclusion is correct I would like to suggest a more detailed explanation about the link between SST and vertical velocity, e.g.

Stronger winds in Arabian Sea => enhanced upwelling along coastline => enhanced cooling of SST => cooler SST advected to central Arabian Sea (via Ekman) where stronger winds cause enhanced downwelling.

Correlation: in central Arabian Sea a negative trend in SST correlates positively with a negative trend in upwelling (i.e. stronger downwelling).

3. Page 7, second and third paragraph: you state that (lines 6 &7) " . . . the identified upwelling trends can presumably be attributed to orbital forcing". Could you please elucidate the known impacts of changes in orbital forcing in Earth System models in the Arabian Sea and reference publication(s) (also see interactive comment by Sebastian Luening).

Last sentence in third paragraph: ". . . the reduction of the SW wind-stress which results from the long-term change of SLP contrast between mid and low latitudes due to the orbital forcing." As above, please add reference(s) regarding the impact of orbital forcing.

Furthermore, did you perform model simulations without the orbital forcing to explore its impact on the trends in SST and wind stress in the Arabian Sea?

4. Page 7, last two lines: ". . . but they rise to similar values by the end of the simulations so the SST modelled by the MPI-ESM-LR has a larger trend than the CCSM."

5. Page 9, lines 22: Arabian Sea

6. Page 9, line 27: ". . . upwelling responds to the change . . ."
* * *

---

## Referee Comment (RC2) · Anonymous Referee #2 · 20 Jan 2017

The study by Yi and Zorita analyzes the temporal evolution of Arabian Sea upwelling over the last millennium and in future climate scenarios using available coupled GCM simulations. Although the topic is potentially of interest to the climate research community, there are several major problems with this study which prevent me from recommending publication of this manuscript.

1) Design of the study: The authors claim that they analyze "the outputs of two earth system models" to study the last millennium and the future: MPI-ESM and CESM. This is not true. In fact they used "MPI-ESM-P" and CESM1 (with CAM5 atmosphere) for the last millennium and "MPI-ESM-LR" and CCSM4 (with CAM4 atmosphere) for the future scenarios. I don't know how much "MPI-ESM-P" differs from "MPI-ESM-

LR" and unfortunately the authors do not explain the difference between these two models. However, it is very clear that CCSM4 and CESM1 are in fact quite different models producing different climates and different climate variability. In other words, the authors used FOUR different models for this study, but a detailed explanation on how these models differ (MPI-ESM-P vs MPI-ESM-LR and CESM1 vs CCSM4) is missing. With that said, I do not understand the intention of analyzing past and future changes in one paper. What is the link between the last millennium and the future projections in this study? Is the intention to validate the models used for projections by means of the last millennium simulations? Given the differences of the models used for the paleo-simulations and the projections, this wouldn't make sense however. In addition, more proxy records than just the G.bulloides record would be needed for a credible validation. Or is the intention to eludicate differences in the forcings between past and future upwelling changes? If this would be the case, then the forcings should be analyzed in much more detail (see my next point).

2) Forcings: The paper does not describe which forcings were included in the last millennium simulations. Much more explanation of the experimental design would be required in the methods section to clarify what drives the externally forced climate variability in the models. Moreover, the study does not properly analyze how the models respond to these forcings. In other words, what causes the correlation between modelled upwelling and G.bulloides-derived upwelling which the authors claim to see in Figure 2. Is it the long-term (orbital-forced) trend or higher frequency fluctuations associated with solar and/or volcanic forcing? Statistical analyzes would be required to examine e.g. spectral coherence (and the G.bulloides record should be shown anyway). Last but not least, are the calculated correlations (Fig. 2) statistically significant? A significance test is missing (which takes autocorrelation of the time series into account).

3) Interpretation of the LME results: The authors state that the "identified upwelling trends can presumably be attributed to the orbital forcing". The LME project, the CESM

results have been taken from, also provides individual forcing experiments (e.g. orbital only). Why didn't the authors use these simulations to test their hypothesis? However, first the authors would need to show that the trends are statistically significant. The p-values shown in Figure 1 do not support significant trends in the CESM simulations. So what are we talking about?

4) Interpretation of the RCP2.6 results: The authors "explain" the lack of a significant upwelling trend in the RCP2.6 simulations by "a compensation between the opposing effects of the increase in upwelling favourable winds and the stratification of the water column". However, p-values shown in Figure 8 do not suggest significant trends in wind stress. What is wrong here? Moreover, it is well known that the monsoon circulation behaves very differently in response to future global warming in different models. The authors should use a much bigger number of models from the CMIP5 pool to come to robust conclusions.

5) "Flip" around year 1550: I don't see a "flip". No appropriate statistics is provided to corroborate a flip. And what should be the forcing of such a flip?

---

## Author Comment (AC1) · 1 Feb 2017

We thank Dr. Sebastian Lüning for his interest in this manuscript and for sharing the detailed comments and the constructive suggestions. In the following, we sketch how we plan to address the two issues brought out by Dr. Sebastian Lüning in the revision.

1. The models used in this study are forced by all forcings including solar forcing, orbital forcing, greenhouse gas forcing, volcanic forcing, etc. So the answer to the first question is: yes, the models account for solar forcing. However, as also mentioned by the two anonymous referees, we agree that we need to explain more clearly the different impacts of solar forcing and orbital forcing on the Arabian Sea upwelling. We need to remark that the Earth System Models are not "calibrated" against the reconstructed

solar irradiance. The external forcings, derived from ice-core records, are independent of the earth system models. There is no model calibration involved regarding the external forcings.

In addition, we thank Dr. Sebastian Lüning for recommending the related literature, which would help us to revise the manuscript.

2. Since our focus is on the last 1000 years, it is not clear to us why a longer proxy record is preferable. It is more suitable to use the data in Anderson et al. 2002, which contains records for approximately the same period. Also, the earth system model simulations that we analyze in our study only run for the last millennium. The proxy record used in Anderson et al. 2010 is indeed improved and extended, but as mentioned in this paper, the maximum during 1200-800 a BP, the minimum at 400 a BP, and the following positive trend were also found in their previous study (Anderson et al. 2002) as well. Therefore, during the last millennium, the updated study shows consistent patterns as the previous one. We thank Dr. Sebastian Lüning for sharing this point and in the revision we will also discuss the updated work of Anderson et al.

---

## Author Comment (AC2) · 1 Feb 2017

We thank the reviewer for the detailed reading of the manuscript and for the suggestions for improvement. In the following, we sketch how we plan to eventually revise this manuscript to address these suggestions.

1. We thank the reviewer for correcting the mistake. This sentence will be changed into "...which is four times coarser than the CCSM4 (288×192)."

2. As suggested by the reviewer, this paragraph will be rewritten by elaborating the link between the SST and vertical velocity. More details will be provided.

3. We are aware of the issue between the orbital forcing and the solar forcing, as mentioned by both reviewers and also by Dr. Sebastian Lüning in the short comment. This is already indirectly included in the manuscript, when we discuss the Mid-Holocene simulations of Braconnot et al. They discuss the effect of Holocene orbital forcing by looking at the differences between equilibrium Mid-Holocene and pre-industrial simulations, but in the revision of the manuscript, we will be more explicit on how the orbital forcing and the solar forcing could affect the model results.

The simulations we analyzed in this study are from the CMIP5 project and from the Last Millennium Ensemble. In those model ensembles there are no millennium simulations excluding the orbital forcing. However, the effect of the orbital forcing, as explained above, can be ascertained with simulations driven by orbital forcing alone. However, since all the simulations present consistent long-term trend and the orbital forcing is the only external forcing that display a millennium scale trend, the long term variation of the modelled upwelling should be connected to the orbital forcing.

4. This sentence will be changed as suggested by the reviewer.

5. We thank the reviewer for pointing out the typo and it will be corrected.

6. Correction will be made as the reviewer suggested.

---

## Author Comment (AC3) · 1 Feb 2017

We thank the reviewer for the detailed reading of the manuscript and for pointing out the problems. In the following, we sketch how we plan to eventually revise this manuscript to address these issues.

1. We agree with the reviewer that a more detailed explanation of the differences among the models is missing. We will improve this by adding a more detailed comparison of the models in the revision of the manuscript. As for the different atmosphere used by CESM1 (CAM5) CCSM4 (CAM4), we could check the results of the CESM1 Large Ensemble simulations, which is also with CAM5 atmosphere, however only RCP8.5 is available.

The objective of the work is to study the impact of external forcing on Arabian Sea upwelling. These external forcings are different over the past millennium and in the future, both in character and amplitude of variations. The issue of model validation is not central, since there are very few instrumental or proxy "observations" or indicators of upwelling. This is why we analyze two (or four, as the reviewer indicated) models and look of their identified responses of upwelling to forcing, or lack therefore, can be explained in a physically consistent way. For instance, we address the question if the knowledge of the upwelling response to forcing over the past millennium can give insights about its response in the future. The conclusion from our study is negative. We find that, in the past, the decadally varying forcings, like solar irradiance or volcanism, do not exert a discernible influence on upwelling, according to the model simulations. In contrast, all simulations do show a consistent response to orbital forcing, which is also consistent with Mid-Holocene simulations, and consistent with the trend in wind in both millennium and Holocene simulations. At decadal time-scales, therefore, upwelling variability is almost purely due to internal variations.

The response to strong scenario future forcing is qualitatively different from the response to orbital forcing. Greenhouse gases cause an intensification of the wind, but this intensification is overridden by the stronger stratification of the water column.

2. The external forcings used in the CMIP5 simulations are explained in many other previous papers, but we understand that problems about the forcings might arise as also mentioned by reviewer #1 and by Dr. Sebastian Lüning in the short comment. In the revision of the manuscript we will explain the differences in the forcings and their impacts on the Arabian Sea upwelling.

We are aware that the significance test is missing for Fig.2. However, in fact all the simulations show similar correlation patterns, although not all of them are significant at the 95% level. We will also explore this when revising the manuscript.

3. We thank the reviewer for the suggestion of analyzing the individual forcing experiments of CESM. However, this is already included in the study, albeit indirectly. The simulated upwelling in all three simulations in each ensemble are uncorrelated with each other at decadal timescales, which means the influences of the forcings varying at decadal timescales, solar variability and clustering of volcanic eruptions, are very small. If their impacts were strong, the upwelling simulated in all simulations will tend to be synchronized. This is not seen in the simulations. The analysis suggested by the reviewer, for instance the spectral coherence between upwelling and forcing, would then be inconclusive. It may well happen that both records, forcing and upwelling, show the same spectral peaks, but the coherence would then be inconsistent among the three simulations, since the simulated upwelling is not synchronized in the simulations. The revised version of the manuscript will explain this point in more detail, although it is already mentioned in the initial submission.

Although the p-values do not support significant trends at 95% level in all simulations, the overall picture is highly statistically significant, even if none of the trends would be, taken in isolation, statistically significant. The reason is that all six simulations show negative trends. Given that all six are independent samples, the probability that this happens by chance is $(0.5)^{**}6 = 0.01$. Therefore, the overall trend analysis, only considering the sign of the trends, is statistically significant at the 0.01. Since the trends are not only all negative, but most are also statistically significant at the individual 0.05 level, the overall significance is much stronger than p=0.01. In other words, it is very unlikely that all six trends are negative, just by chance. We would explain this point in more detail in a revised version.

The reviewer also suggests to use the CESM simulations forced by orbital forcing. In this study we have looked at the orbital forcing in a slightly different way, as the difference between the mid-Holocene and pre-industrial equilibrium simulations, which provide the same circulation trends over Asia as the millennium simulations. In the revised version we will confirm these results by looking at the upwelling trends simulated in three orbital only simulations of the LME with the model CESM.

4. We do not completely understand the reviewer's concern about the missing interpretation for the lack of a significant upwelling trend in RCP2.6 as the compensation between the increasing winds and the water stratification. Quite likely, this is due to our unfortunate formulation of this paragraph. The externally forced trends in wind and upwelling will likely be the same signs as in the strongest scenario RCP8.5, although in the weaker scenario those trends may be smaller and therefore not significant. This does not mean that they do not exist. Our interpretation is that in the weaker scenario, both effects are in a closer position to compensate each other, in contrast to the strongest scenario where the effect of stratification dominates. However, this is a possible (likely) interpretation. If, for instance, the imprint of the external GHG forcing on wind were non-linear and changed sign with the strength of the forcing, this interpretation would be invalid.

We thank the reviewer for the suggestion to analyze the changes in future Monsoon circulation in more CMIP5 models. That analysis has been already conducted in other context with focus on changes in Indian rainfall (Menon et al., 2013, doi:10.5194/esd-4-287-2013). These authors actually found that the simulated changes in the Monsoon circulation are quite robust across the CMIP model ensemble and consistent with the changes simulated by the MPI-ESM and CESM models, i.e. a strengthening of the upwelling favorable winds in the future under RCP8.5 scenario. The revised manuscript would incorporate a more detailed discussion of this point.

5. The change in trend in the G.Bulloides record was reported by Anderson et al (2002). We agree with the reviewer that the flip in the model simulations is not as clear as in the G.Bulloides record and it should be better supported in the manuscript. More statistical evidence will be provided in the revision. A possible cause of the change in trend might be related to the solar irradiance forcing, as there is a multi-centennial trend from about 1700 to present in the solar forcing. If the statistical analysis fails to detect a change in trend in the models, the revised version will document the disagreement with the G.Bulloides record on this point.